# Fault estimation for T-S fuzzy systems via an $L_\infty$ switching observer scheme

1st Yue Wu
*School of Electrical Engineering*
*Southwest Jiaotong University*
Chendu, China
wy2020@swjtu.edu.cn

2nd Kai Zhang
*School of Aeronautics and Astronautics*
*Sichuan University*
Chengdu, China
zhangkaihit@gmail.com

3rd Yang Wang
*Navigation College*
*Dalian Maritime University*
Dalian, China
wangyangyouth@163.com

4th Xiaojie Sun
*College of Information Engineering*
*Henan University of Science and Technology*
Luoyang, China
xjsun.phd@gmail.com

5th Shanfeng Zhang
*School of Electrical Engineering*
*Southwest Jiaotong University*
Chendu, China
2019112113@my.swjtu.edu.cn

*Abstract*—This paper addresses the fault estimation (FE) problem for T-S fuzzy systems based on the switching observer scheme. Firstly, a fuzzy FE observer with switching rules is designed, where the product of time derivative of the membership functions and Lyapunov matrix can be guaranteed to be negative definite. Furthermore, to increase the robustness of the observer against the persistence disturbance, an $L_\infty$ performance index is considered for synthesizing the FE observer. Then, based on the Lyapunov stability analysis method and the average dwell-time technique, the synthesis conditions of the $L_\infty$ FE switching observer are given in terms of linear matrix inequalities (LMIs). The merit of the proposed method is that by using the switching rules, the constraint in the conventional methods, where the state should be limited into a local area, can be removed. Meanwhile, based on the $L_\infty$ performance analysis, the developed approach has a good FE performance for the systems with persistent disturbance. Finally, an example is given to show the efficacy of the presented FE scheme.

*Index Terms*—T-S fuzzy systems, fault estimation, switching observer design, $L_\infty$ performance.

## I. INTRODUCTION

Fault diagnose (FD) technology is an effective tool to ensure the reliability and safety of the control systems. As a research branch of the FD method, the fault estimation technology can provide exact information of the fault, such as, the magnitude or size of the fault, which is helpful for improving the effect of FD. During the past years, the FE problem has received considerable attentions and many interesting methods on this topic have been reported in [1]- [3]. Among all the methods, the FE method based on T-S fuzzy model has become a hot research topic due to the advantage of T-S fuzzy system in dealing with the nonlinear problem. In recent years, many researchers have devoted themselves to solving the FE problem of T-S fuzzy systems and good FE performance for the nonlinear systems have been achieved [5]- [7].

However, there is still some space to improve the design. In some existing methods, the fuzzy Lyapunov function (FLF) is used to analyze the stability of the systems, which can decrease the conservatism of the design. However, when using FLF, the derivatives of the membership functions will appear in the Lyapunov inequalities which makes the the synthesis conditions of the observer non-convex. In order to avoid coping with the derivatives of FLFs, the constraint where the state of the systems should be restricted into a local area is imposed in some literatures. This limits the application range of the methods. To solve the problem, a new switching observer scheme is proposed in [8] to estimate the state of the system. By utilizing the switching rules, there is no need to handle with the derivative of the membership functions in the synthesis of the fuzzy observer. Inspired by the [8], the design of the switching FE observer scheme is studied in this paper to remove the above-mentioned constraints.

Another interesting research area is the $H_\infty$ observer design. The $H_\infty$ technology can provide an effective way to decrease the influence of the disturbance on the estimated error of the observer such that it has been widely used to solve the design problem of the robust observer [9]- [11]. Under the framework of the $H_\infty$ observer schemes, there is an important assumption where the disturbance considered should be energy-bounded. However, in real applications, the disturbance signals to which the systems are subjected are usually persistent rather than energy-bounded. To make the method more applicable for the persistent disturbance, we resort to an alternative method, i.e., $L_\infty$ analysis method, so as to present the FE observer scheme.

Observing the afore-mentioned points, the design problem

This work was supported by the Opening Project of Robotic Satellite Key Laboratory of Sichuan Province, the Joint Fund of Science and Technology Department of Liaoning Province and State Key Laboratory of Robotics under Grant 2022-KF-22-08, the Key Scientific Research Projects for Colleges and Universities of Henan Province of China under Grant 23A413006, the Science and Technology Research Project of Henan Province of China under Grant 242102221056 and the 74th batch of General Projects Funded by the China Postdoctoral Science Foundation under Grant 2023M741038.

of $L_\infty$ switched FE observer scheme is investigated in this paper. The contributions of this paper are listed as follows:

(1) Motivated by [8], an FE scheme with the switching rules is proposed in this paper. By using the switching rules, the difficulty in solving LMIs introduced by fuzzy Lyapunov functions can be overcome. In addition, different from [8], the membership functions in the proposed switching rules depend on state estimations such that the proposed method can be applicable for the system with unmeasurable premise variables.

(2) Based on the average dwell-time technique, the synthesis conditions for the switching $L_\infty$ FD observer are obtained such that the robustness of the developed method against the persistent disturbance can be increased.

The paper is organized as follows: In Section II, the T-S fuzzy systems under consideration are described and the $L_\infty$ switching FE observer strategy is presented. The sufficient condition of the existence of the FE observer is given in Section III. An example with respect to the mass-spring-damper system is provided in Section IV to demonstrate the advantage of the developed FE observer scheme. Section V draws the conclusion.

**Notation**: $\mathbb{R}^n$ is the $n$-dimensional Euclidean space. $P > 0$ denotes that the matrix $P$ is positive definite. $||y(t)||_\infty = \sup_{t \geq 0} ||y(t)||_2$ stands for the $L_\infty$ norm of a signal $y(t)$, where $||y(t)||_2 = \sqrt{y^T(t)y(t)}$. For a matrix $X$, $\lambda_{min}(X)$ and $\lambda_{max}(X)$ represent its smallest and largest eigenvalues, respectively.

## II. PROBLEM STATEMENT AND PRELIMINARIES

### A. System description

The nonlinear systems under consideration are approximated by the following T-S fuzzy systems:

**Plant Rule $i$:**

IF $\xi_1(t)$ is $\mathcal{M}_{i1}$ and $\xi_2(t)$ is $\mathcal{M}_{i2}, \cdots, \xi_p(t)$ is $\mathcal{M}_{ip}$
THEN $\dot{x}(t) = A_i x(t) + B_i u(t) + F_i f(t) + D_i \omega(t)$
$$y(t) = Cx(t) \tag{1}$$

where $x(t) \in \mathbb{R}^{n_x}$ is the state of the system and $u(t) \in \mathbb{R}^{n_u}$ represents the system input; $\omega(t) \in \mathbb{R}^{n_\omega}$ and $f(t) \in \mathbb{R}^{n_f}$ denote the disturbance and fault signals, respectively; $\xi(t) = [\xi_1(t), \xi_2(t), \ldots, \xi_p(t)]$ is the vector of the premise variables which are assumed to be unmeasurable; $\mathcal{M}_{ij}(i = 1, \cdots, r, j = 1, \cdots, p)$ represent the fuzzy sets; $r$ and $p$ denote the number of the fuzzy rules and the premise variables, respectively; $A_i$, $B_i$, $F_i$ $D_i$ and $C$ are the real constant matrices with compatible dimensions.

By using the standard modeling process, the global model of the system (1) is given as follows:

$$\dot{x}(t) = \sum_{i=1}^{r} h_i(\xi(t)) [A_i x(t) + B_i u(t) + F_i f(t) + D_i \omega(t)]$$
$$y(t) = Cx(t) \tag{2}$$

where $h_i(\xi(t)) = \frac{\mu_i(\xi(t))}{\sum_{i=1}^{r} \mu_i(\xi(t))}$ and $\mu_i(\xi(t)) = \prod_{j=1}^{p} \mathcal{M}_{ij}(\xi_j(t))$.
Assume that $\mu_i(\xi(t)) \geq 0, i = 1, \cdots, r$, then we have

$$\sum_{i=1}^{r} h_i(\xi(t)) = 1 \text{ and } h_i(\xi(t)) \geq 0. \tag{3}$$

To obtain the switch rule of the FE observer, an important property of $\dot{h}(\xi(t))$ is discussed here. In light of the condition (3), one can obtain the following property.

$$\sum_{i=1}^{r} \dot{h}_i(\xi(t)) = 0. \tag{4}$$

### B. The scheme of the FE observer

Before introducing the FE observer scheme, the following output feedback controller is considered to stabilize the system (2)

$$u(t) = Ky(t) \tag{5}$$

Then, substituting the control law (5) into (2), we can obtain

$$\dot{x}(t) = \sum_{i=1}^{r} h_i(\xi(t)) [\bar{A}_i x(t) + F_i f(t) + D_i \omega(t)]$$
$$y(t) = Cx(t) \tag{6}$$

where $\bar{A}_i = A_i + B_i KC$.

**Remark 1**. *This paper mainly concerns the problem of FE observer design rather than the design of the controller. Thus, the controller is assumed to have been synthesized by designer in advance. As for how to design an effective controller scheme, the readers can refer to some effective control literatures such as [12]- [13].*

Next, a switching-type FE observer is designed for the system (6):

$$\dot{\hat{x}}(t) = \sum_{i=1}^{r} h_i(\hat{\xi}(t)) [\bar{A}_i \hat{x}(t) + L_i^q (y(t) - \hat{y}(t)) + F_i \hat{f}(t)]$$
$$\hat{y}(t) = C\hat{x}(t)$$
$$\dot{\hat{f}}(t) = \sum_{i=1}^{r} h_i(\hat{\xi}(t)) H_i^q (y(t) - \hat{y}(t)) \tag{7}$$

where $\hat{x}(t)$, $\hat{y}(t)$ and $\hat{f}(t)$ represent the estimations of the state, output and fault signal, respectively. $H_i^q$ and $L_i^q, (i = 1, \cdots, r, q = 1, \cdots, 2^{r-1})$ are switching observer gain matrices which are determined later.

**Remark 2**. *In this paper, the premise variables $\xi(t)$ are assumed to depend on the unmeasurable state. As a consequence, to achieve the FE observer design, the premise variables of the observer (7) depend on $\hat{x}(t)$ instead of $x(t)$.*

Based on the property (4) and the dynamic (7), the switching rule of the FE observer is given in Lemma 1.

**Lemma 1 [8].** *For any membership function-dependent matrices $\dot{P}_{\hat{h}}$, where $\dot{P}_{\hat{h}} = \sum_{i=1}^{r} \dot{h}_i(\hat{\xi}(t)) P_i$ and $P_i > 0$, the following*

*conditions hold*

$$\dot{P}_{\hat{h}} = \sum_{i=1}^{r-1} \dot{h}_i\big(\hat{\xi}(t)\big)P_i + \dot{h}_r\big(\hat{\xi}(t)\big)P_r$$

$$= \sum_{i=1}^{r-1} \dot{h}_i\big(\hat{\xi}(t)\big)\big(P_i - P_r\big) < 0 \qquad (8)$$

*if the following switch rule is satisfied for $\forall i \in \{1, \ldots, r\}$*

$$\begin{cases} \text{If } \dot{h}_i\big(\hat{\xi}(t)\big) \leq 0, \text{ Then } P_i - P_r > 0, \\ \text{If } \dot{h}_i\big(\hat{\xi}(t)\big) > 0, \text{ Then } P_i - P_r < 0. \end{cases} \qquad (9)$$

**Remark 3.** From Lemma 1, it is clear that $\dot{P}_{\hat{h}} < 0$ can be guaranteed by the switching rule (9). A more detailed discussion about the switching rule could be seen in [8]. Besides, it should be pointed out that there exists a difference between the proposed switching rule (9) and the one in [8]. In [8], the switching rule depends on measurable premise variables. However, in this paper, the premise variables (PVs) are assumed to be immeasurable so that PVs in the switching rule (9) depends on $\hat{x}(t)$.

Define the state estimation error and fault estimation error as $e_x(t) = x(t) - \hat{x}(t)$ and $e_f(t) = f(t) - \hat{f}(t)$. Then, combined with (6) and (7), the estimation error dynamics can be obtained as follows:

$$\dot{\tilde{x}}(t) = \sum_{i=1}^{r}\sum_{j=1}^{r} h_i(t)\hat{h}_j(t)\big[\widetilde{A}_{ij}\tilde{x}(t) - \widetilde{L}_j^q\bar{C}\tilde{x}(t) + \widetilde{D}_i\bar{\omega}(t)\big]$$

$$(10)$$

where $\tilde{x}(t) = [x^T(t) \ e_x^T(t) \ e_f^T(t)]^T$, $\bar{C} = [C \ 0 \ 0]$, $\widetilde{L}_j^q = \big[0 \ (L_j^q)^T \ (H_j^q)^T\big]^T$, $\bar{\omega}(t) = \big[\omega^T(t) \ f^T(t) \ \dot{f}^T(t)\big]^T$,

$$\widetilde{A}_{ij} = \begin{bmatrix} A_i + B_i K C & 0 & 0 \\ A_i - A_j & A_j & F_i \\ 0 & 0 & 0 \end{bmatrix}, \widetilde{D}_i = \begin{bmatrix} D_i & F_i & 0 \\ D_i & 0 & 0 \\ 0 & 0 & I \end{bmatrix},$$

$h_i(t) = h_i\big(\xi(t)\big), \hat{h}_i(t) = h_i\big(\hat{\xi}(t)\big)$ for brevity.

In the following, two useful definitions are given for obtaining the main result.

**Definition 1.** ( [14]) *Under the zero initial condition, i.e., $\tilde{x}(t_0) = 0$, the system has an $L_\infty$ performance index bound $\gamma$ if the following inequality is satisfied*

$$||\tilde{x}(t)|| \leq \bar{\gamma}||\tilde{\omega}(t)||_\infty. \qquad (11)$$

**Definition 2.** ( [15]) *For the switched system (10), if the following condition holds in the interval $[t_1, t_2]$*

$$N(t_1, t_2) \leq N_0 + \frac{t_2 - t_1}{T_a} \qquad (12)$$

*then, $T_a$ is said to be the average dwell time, where $N(t_1, t_2)$ represents the number of switching in the time interval $[t_1, t_2]$ and $N_0$ denotes chattering bound which is generally assumed to be 0.*

The main problem of the this paper is formulated as: to find the switching FE observer gain matrices $H_i^q$ and $L_i^q$ in

(7) such that the error system (10) satisfies the following two requirements:

(1) When $\tilde{\omega}(t) = 0$, the system (10) is exponentially asymptotically stable.

(2) When $\tilde{\omega}(t) \neq 0$, the system (10) satisfies $L_\infty$ performance index (11).

## III. MAIN RESULTS

In this section, an $L_\infty$ FE observer scheme is presented to solve the problem formulated in Section II and the corresponding synthesis conditions of the FE observer are given in Theorem 1.

**Theorem 1.** *For given positive scalars $\alpha$, $\beta$, $\bar{\theta}$, $\varepsilon_1$ and $\varepsilon_2$, if there exist matrices $P_j^k > 0$, $X$ and $Q_j^k$ such that the following inequalities hold for $\forall i = 1, \ldots, r, j = 1, \ldots, r, k, m = 1, \ldots, 2^r - 1$*

$$\begin{bmatrix} \Omega_{ij}^k & -\varepsilon_1 X - \varepsilon_1(X\widetilde{A}_{ij} - Q_j^k\bar{C})^T + P_j^k & \varepsilon_1 X\widetilde{D}_i \\ * & -\varepsilon_2 He(X) & \varepsilon_2 X\widetilde{D}_i \\ * & * & -\bar{\theta}I \end{bmatrix} < 0,$$

$$(13)$$

$$P_j^m < \beta P_j^k, \forall m \neq k \qquad (14)$$

$$I < P_j^k \qquad (15)$$

*where $\Omega_{ij}^k = He(X\widetilde{A}_{ij} - Q_j^k\bar{C}) + \alpha P_j^k$ and $\bar{\gamma} = \sqrt{\frac{\bar{\theta}}{\frac{\ln\beta}{T_a} - \alpha}}$, then, the error system (10) is exponentially asymptotically stable with $L_\infty$ performance index bound $\bar{\gamma}$ for arbitrary switching signal with average dwell time $T_a > \frac{\ln\alpha}{\beta}$. Moreover, the matrices $\widetilde{L}_i^k$ can be calculated by $\widetilde{L}_i^k = X^{-1}Q_j^k$.*

*Proof:* Select the Lyapunov functional as

$$V(t) = \tilde{x}^T(t)\sum_{j=1}^{r} \hat{h}_j(t)P_j^k\tilde{x}(t), k = 1, \cdots, 2^{r-1} \qquad (16)$$

Calculating $\dot{V}(t)$ yields

$$\dot{V}(t) = \tilde{x}^T(t)\sum_{j=1}^{r}\hat{h}_j(t)P_j^k\dot{\tilde{x}}(t) + \dot{\tilde{x}}^T(t)\sum_{j=1}^{r}\hat{h}_j(t)P_j^k\tilde{x}(t)$$

$$+ \tilde{x}^T(t)\sum_{j=1}^{r}\dot{\hat{h}}_j(t)P_j^k\tilde{x}(t)$$

$$= \tilde{x}^T(t)\sum_{j=1}^{r}\hat{h}_j(t)P_j^k\dot{\tilde{x}}(t) + \dot{\tilde{x}}^T(t)\sum_{j=1}^{r}\hat{h}_j(t)P_j^k\tilde{x}(t)$$

$$+ \tilde{x}^T(t)\sum_{j=1}^{r-1}\dot{\hat{h}}_j(t)(P_j^k - P_r^k)\tilde{x}(t) \qquad (17)$$

Based on the switching rule (9) and (17), we have

$$\dot{V}(t) \leq \tilde{x}^T(t)\sum_{j=1}^{r}\hat{h}_j(t)P_j^k\dot{\tilde{x}}(t) + \dot{\tilde{x}}^T(t)\sum_{j=1}^{r}\hat{h}_j(t)P_j^k\tilde{x}(t)$$

$$(18)$$

On the other hand, on the basis of the dynamic (7), the following zero equation can be derived

$$\sum_{i=1}^{r}\sum_{j=1}^{r}h_i(t)\hat{h}_j(t)\Big\{2\big[\tilde{x}^T(t)\varepsilon_1 X + \dot{\tilde{x}}^T(t)\varepsilon_2 X\big] \times \big[\widetilde{A}_{ij}\tilde{x}(t)$$
$$- \widetilde{L}_j^k\bar{C}\tilde{x}(t) + \widetilde{D}_i\bar{\omega}(t) - \dot{\tilde{x}}(t)\big]\Big\} = 0 \quad (19)$$

where the given scalars $\varepsilon_1 > 0$ and $\varepsilon_2 > 0$; $X$ is an invertible matrix.

Combining with (18) and (19) and making variable change $Q_j^k = XL_j^k$, one has

$$\dot{V}(t) + \alpha V(t) - \bar{\theta}\bar{\omega}^T(t)\bar{\omega}(t)$$
$$\leq \chi^T(t)\sum_{i=1}^{r}\sum_{j=1}^{r}h_i(t)\hat{h}_j(t)\Xi_{ij}^k\chi(t) \quad (20)$$

where

$$\Xi_{ij}^k = \begin{bmatrix} \Omega_{ij}^k & -\varepsilon_1 X - \varepsilon_1(X\widetilde{A}_{ij} - Q_j^k\bar{C})^T + P_j^k & \varepsilon_1 X\widetilde{D}_i \\ * & -\varepsilon_2 He(X) & \varepsilon_2 X\widetilde{D}_i \\ * & * & -\bar{\theta}I \end{bmatrix}.$$

It can be known from (20) that if the condition (13) holds, then

$$\dot{V}(t) \leq -\alpha V(t) + \bar{\theta}\bar{\omega}^T(t)\bar{\omega}(t). \quad (21)$$

Consider the case $\bar{\omega}(t) = 0$, then, it follows from (21) that

$$\dot{V}(t) \leq -\alpha V(t) \Rightarrow \frac{d(e^{\alpha t}V(t))}{dt} < 0 \quad (22)$$

Integrating the condition (22) from $t_n$ to $t$ gives

$$V(t) < e^{-\alpha(t-t_n)}V(t_n) \quad (23)$$

where $t_n$ denotes the $n$-$th$ switching time; $t \in [t_n, t_{n+1})$.

On the other hand, based on (14), we have

$$V\big(\tilde{x}(t_n)\big) < \beta V\big(\tilde{x}(t_n^-)\big) \quad (24)$$

Taking (23) and (24) into consideration, one can obtain

$$V\big(\tilde{x}(t)\big) < e^{-\alpha(t-t_n)}\beta V\big(\tilde{x}(t_n^-)\big) < e^{-\alpha(t-t_{n-1})}\beta V\big(\tilde{x}(t_{n-1})\big)$$
$$< e^{-\alpha(t-t_{n-1})}\beta^2 V\big(\tilde{x}(t_{n-1}^-)\big) < \cdots$$
$$< e^{-\alpha(t-t_0)}\beta^{N(t_0,t)}V\big(\tilde{x}(t_0)\big) \quad (25)$$

With Definition 2, we have

$$V\big(\tilde{x}(t)\big) < e^{-(t-t_0)\left(\alpha - \frac{ln\beta}{T_a}\right)}V\big(\tilde{x}(t_0)\big). \quad (26)$$

Based on the properties $\bar{\lambda}_{min}(P)||\tilde{x}(t)||^2 < V\big(\tilde{x}(t)\big)$ and $V\big(\tilde{x}(t_0)\big) \leq \bar{\lambda}_{max}(P)||\tilde{x}(t_0)||^2$, one can get

$$\bar{\lambda}_{min}(P)||\tilde{x}(t)||^2 < V\big(\tilde{x}(t)\big) < e^{-(t-t_0)\left(\alpha - \frac{ln\beta}{T_a}\right)}V\big(\tilde{x}(t_0)\big)$$
$$\leq \bar{\lambda}_{max}(P)e^{-(t-t_0)\left(\alpha - \frac{ln\beta}{T_a}\right)}||\tilde{x}(t_0)||^2 \quad (27)$$

where

$$\bar{\lambda}_{min}(P) = min\{\lambda_{min}(P_1^1), \lambda_{min}(P_1^2), \cdots, \lambda_{min}(P_r^{2^r-1})\}$$

and

$$\bar{\lambda}_{max}(P) = max\{\lambda_{max}(P_1^1), \lambda_{max}(P_1^2), \cdots, \lambda_{max}(P_r^{2^r-1})\}.$$

It can be inferred from (27) that the system (10) is exponentially asymptotically stable when $\bar{\omega}(t) = 0$.

Next, we will proof that the error system (10) satisfies the $L_\infty$ performance index (11).

Similar to the procedure of the previous proof, the following condition can be obtained if we have the condition (21).

$$V\big(\tilde{x}(t)\big) < e^{-\alpha(t-t_n)}V\big(\tilde{x}(t_n)\big) + \int_{t_n}^{t}\bar{\theta}\bar{\omega}^T(s)\bar{\omega}(s)ds \quad (28)$$

where $t \in [t_n, t_{n+1})$.

Following the method in [16] and considering the condition (14), one can get

$$V\big(\tilde{x}(t)\big)$$
$$< \beta e^{-\alpha(t-t_n)}V\big(\tilde{x}(t_n^-)\big) + \int_{t_n}^{t}e^{-\alpha(t-s)}\bar{\theta}\bar{\omega}^T(s)\bar{\omega}(s)ds$$
$$< \beta^n e^{-\alpha(t-t_0)}V\big(\tilde{x}(t_0)\big) + \int_{t_n}^{t}e^{-\alpha(t-s)}\bar{\theta}\bar{\omega}^T(s)\bar{\omega}(s)ds+$$
$$\int_{t_{n-1}}^{t_n}\beta e^{-\alpha(t-s)}\bar{\theta}\bar{\omega}^T(s)\bar{\omega}(s)ds + \int_{t_{n-1}}^{t_n}\beta^2 e^{-\alpha(t-s)}\bar{\theta}\bar{\omega}^T(s)$$
$$\bar{\omega}(s)ds + \cdots + \int_{t_0}^{t_1}\beta^n e^{-\alpha(t-s)}\bar{\theta}\bar{\omega}^T(s)\bar{\omega}(s)ds \quad (29)$$

Under the zero initial condition, it follows from (29) that

$$V\big(\tilde{x}(t)\big) < \int_{t_0}^{t}e^{N(s,t)ln\beta - \alpha(t-s)}\bar{\theta}\bar{\omega}^T(s)\bar{\omega}(s)ds$$
$$< \int_{t_0}^{t}e^{\left(\frac{ln\beta}{Ta} - \alpha\right)(t-s)}\bar{\theta}\bar{\omega}^T(s)\bar{\omega}(s)ds$$
$$< \frac{\bar{\theta}}{\frac{ln\beta}{Ta} - \alpha}||\omega(s)||^2 \quad (30)$$

where $N(s,t)$ is the function which take the value $0, 1, \cdots,$ $N(t_0, t)$, respectively, when $s \in [t_n, t), [t_{n-1}, t_n), \ldots, [t_0, t_1)$.

On the other hand, according to (15), we can obtain

$$||\tilde{x}(t)||^2 < \tilde{x}^T(t)\left(\sum_{i=1}^{r}\hat{h}_i P_i^k\right)\tilde{x}(t) = V\big(\tilde{x}(t)\big) \quad (31)$$

By virtue of (30) and (31), one has

$$||\tilde{x}(t)|| < \sqrt{\frac{\bar{\theta}}{\frac{ln\beta}{Ta} - \alpha}}||\omega(s)|| < \sqrt{\frac{\bar{\theta}}{\frac{ln\beta}{Ta} - \alpha}}||\omega(s)||_\infty \quad (32)$$

Here, the proof is completed. ∎

## IV. EXAMPLE

In this paper, an example with respect to a mass-spring-damper system is given to show the effectiveness of the presented approach. The dynamic of the mass-spring-damper system considered in this paper is given as follows:

$$M\ddot{s} + g(s,\dot{s}) + q(s) + \varphi_1(s)d = \varphi_2(s)u \quad (33)$$

where $M$ is the mass; $u$ is the force of the system; The displacement of the system is represented by $s$; $\varphi_1(t)$, $\varphi_2(t)$, $g(s,\dot{s})$ and $q(s)$ are the nonlinear functions with respect to disturbance, input, damper and spring, respectively.

In this example, suppose that $\varphi_1(s) = 0.2$, $\varphi_2(s) = 1$, $M = 1$, $g(s, \dot{s}) = -0.75\dot{s}$, $q(s) = 0.67s^3 - 0.05s$.

Next, select the state variables as $x_1(t) = s$ and $x_2(t) = \dot{s}$ and using the T-S fuzzy system to represent the dynamic (33) on the compact set $S = \{x(t)\big|\,|x_1(t)| < 1.5\}$, we can get

$$\dot{x}(t) = \sum_{i=1}^{2} h_i(t)\big[A_i x(t) + B_i u(t) + F_i f(t) + D_i \omega(t)\big]$$
$$y(t) = Cx(t) \tag{34}$$

where $h_1(t) = 1 - \frac{x_1^2(t)}{2.25}$, $h_2(t) = 1 - h_1(t)$, $C = \begin{bmatrix} 1.05 & 2.75 \end{bmatrix}$,

$$A_1 = \begin{bmatrix} 0 & 1 \\ 0.05 & 0.75 \end{bmatrix}, A_2 = \begin{bmatrix} 0 & 1 \\ -1.4575 & 0.75 \end{bmatrix}, B_1 = \begin{bmatrix} 0 \\ 1 \end{bmatrix},$$

$$B_2 = \begin{bmatrix} 0 \\ 1 \end{bmatrix}, D_1 = D_2 = \begin{bmatrix} 0 \\ 0.2 \end{bmatrix}, F_1 = F_2 = \begin{bmatrix} 0 \\ 1 \end{bmatrix}.$$

Based on the output matrix $C$ in (34), we know that the premise variables $h_1(t)$ and $h_2(t)$ are unmeasurable. Thus, the method in [17] is not applicable for this example, while our method can be utilized to deal with the FE problem for the T-S system (34) with immeasurable premise variables.

Assume that the output feedback controller $u(t)$ has been designed in advance and the control gain matrix $K = -2$.

Then, we can get the following closed-loop system

$$\dot{x}(t) = \sum_{i=1}^{r} h_i(\xi(t))\big[\bar{A}_i x(t) + F_i f(t) + D_i \omega(t)\big]$$
$$y(t) = Cx(t) \tag{35}$$

where

$$\bar{A}_1 = \begin{bmatrix} 0 & 1.00 \\ -2.05 & -4.75 \end{bmatrix}, \bar{A}_2 = \begin{bmatrix} 0 & 1.00 \\ -3.5575 & -4.75 \end{bmatrix}.$$

Set $\bar{\theta} = \sqrt{2}$, $\varepsilon_1 = 20$, $\varepsilon_2 = 4$, $\beta = 2$ and $\alpha = 0.1$, then, by solving the LMIs in Theorem 1, the switching FE observer gain matrices can be obtained as:

$$L_1^1 = \begin{bmatrix} 0.434 \\ 16.622 \end{bmatrix}, L_1^2 = \begin{bmatrix} 0.426 \\ 15.091 \end{bmatrix}, H_1^1 = 91.490, H_1^2 = 83.601,$$

$$L_2^1 = \begin{bmatrix} 0.405 \\ 16.181 \end{bmatrix}, L_2^2 = \begin{bmatrix} 0.436 \\ 16.382 \end{bmatrix}, H_2^1 = 91.976, H_2^2 = 92.552$$

With the initial condition $x_0 = [-0.2, 0.4]$ and suppose that the disturbance signal to which the system is subjected is $d(t) = 0.1 sin(t)$.

In addition, two cases of the fault signal are considered to verify the effectiveness of the proposed method.

**Case 1:**

$$f(t) = \begin{cases} 0, & 0 \le t \le 30, \\ 0.1(t - 30), & 30 < t < 70, \\ 4, & t \ge 70. \end{cases}$$

**Case 2:** $f(t) = sin(0.2t)$.

The simulation results of FE for the fault in Case 1 are shown in Figs. 1-3. Figs. 1-2 show the system state, fault signal and their estimations, respectively. From Fig.1, it can be seen

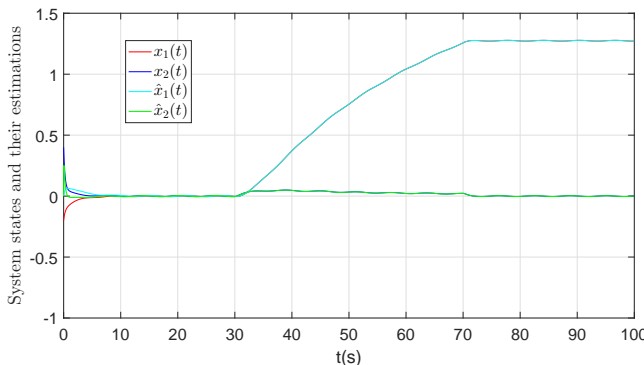

Fig. 1. The states and their estimations for case 1.

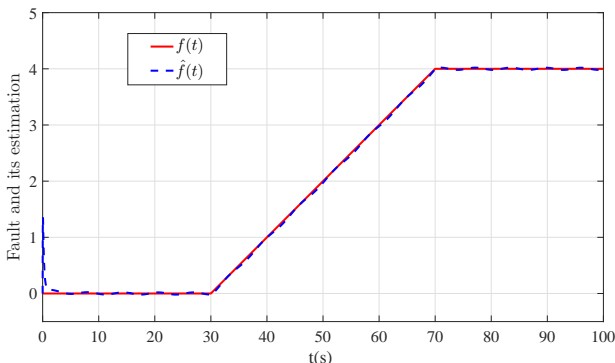

Fig. 2. The fault and its estimations for case 1.

that the proposed method has a good state estimation and FE performance. Fig. 3 depicts the fault estimation error generated by our method and the FE method without $L_\infty$ performance analysis. We can see from Fig. 3 that compared with the method without $L_\infty$ performance analysis, the developed method can get a smaller fault estimation error. It indicates that the $L_\infty$ performance analysis is helpful for increasing the robustness of the observer against the persistent disturbance.

The simulation results of FE for the fault in Case 2 are shown in Figs. 4-6. Figs. 4-5 plot the trajectories of the

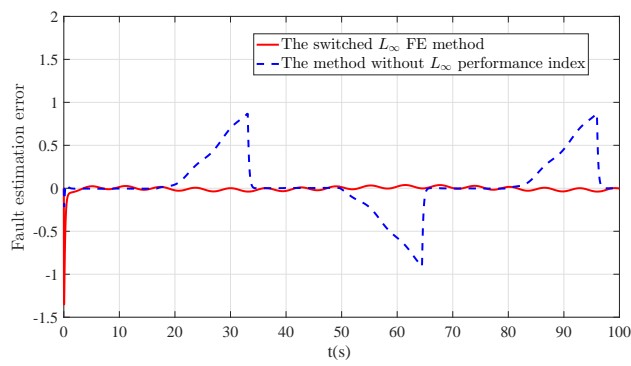

Fig. 3. The error of fault estimation.

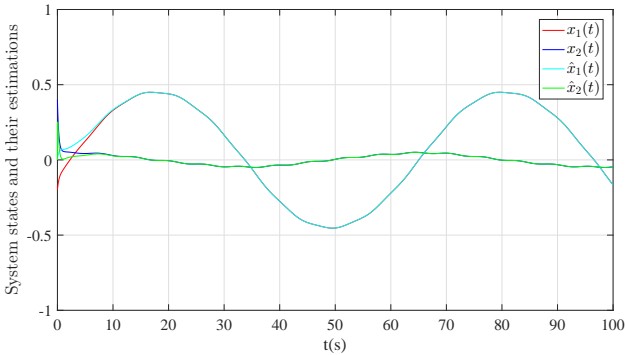

Fig. 4. The states and their estimations for case 2.

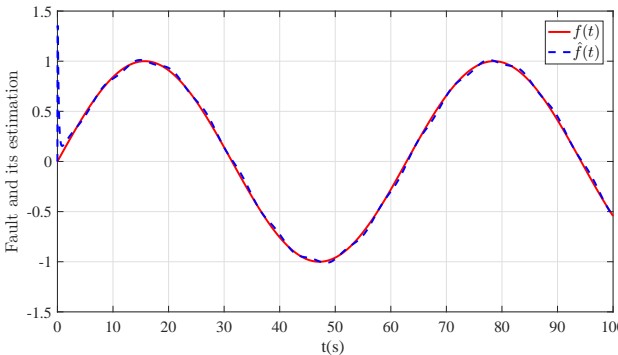

Fig. 5. The fault and its estimations for case 2.

state and FE estimations and their estimations. Fig. 6 gives the comparison results between the presented method and the approach without $L_\infty$ performance index. From these figures, it is easy to see that the proposed method can well estimate the state and fault of the system.

## V. CONCLUSION

This paper studies with the FE problem for T-S fuzzy systems with unmeasurable premise variables. Firstly, T-S fuzzy systems are used to represent the nonlinear systems. Secondly, by considering the controller which is designed

in advance, a closed-loop system is established. Based on the closed-loop system, a switching-type $L_\infty$ FE observer is designed. Thirdly, a sufficient condition for the existence of the FE observer is given in terms of LMIs. In the end, the effectiveness of the developed system is verified by the example. In the future, we will concentrate on the filtering problem for T-S fuzzy system via the switching strategy.

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

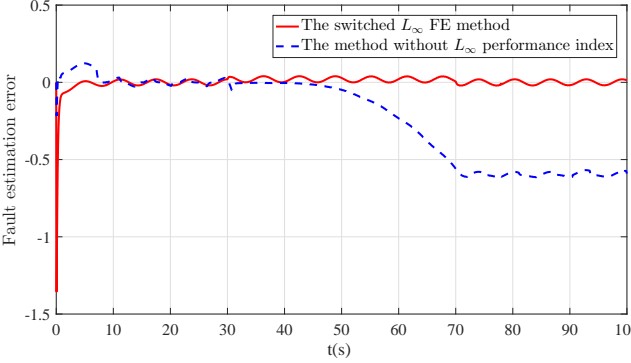

Fig. 6. The error of fault estimation.