# OpenReview forum: "Fault estimation for T-S fuzzy systems via an $L_\infty$ switching observer scheme"
_IEEE.org/ICIST/2024/Conference — IEEE ICIST 2024 Conference Submission_

### Official Review · Reviewer_chYz · 2024-08-21
**Good work**

**Rating:** 7
**Confidence:** 3

**Review:**

The paper titled "Fault estimation for T-S fuzzy systems via an  $L_\infty$ switching observer scheme" proposes a switching observer for T-S fuzzy systems. In the theoretical part, a fuzzy FE observer with switching rules is designed. And its $L_\infty$  performance index is investigated. My specific feedback is as follows:
1) Some formatting issues need to be addressed.
2) The motivation for studying switching observers can be illustrated more clearly in Introduction.

---

### Official Review · Reviewer_RJay · 2024-08-21
**Accept**

**Rating:** 7
**Confidence:** 3

**Review:**

This paper studied with the FE problem for T-S fuzzy systems with unmeasurable premise variables. The theory is correct and can be accepted after responding the following comments.
(1)	In the introduction, it is not enough to state the current work. It should be expended and reconstructed.
(2)	There are many typos and grammar errors. The authors should have a native English speaker or software packages to perform the editing check.
(3)	The conclusion of the article suggests using the present perfect tense for description.

---

### Official Review · Reviewer_ijoC · 2024-08-28
**Fault estimation for T-S fuzzy systems via an   switching observer scheme**

**Rating:** 7
**Confidence:** 2

**Review:**

This paper addresses the fault estimation (FE) problem for T-S fuzzy systems based on the switching observer scheme.
a The abstract should mainly include elements such as research purpose, methods and final results, and reviewers suggest optimizing the content of the abstract.
b What are the significant differences between this study and previous studies? The author needs more explicit emphasis.
c There are some grammatical mistakes and typos. Please examine the full text further and revise them.
d The references should be updated. Some closely related and new references should be added to show to further explain the novelty and innovation of the work.

---

### Decision · Program_Chairs · 2024-09-06

Accept (Oral)